# The Effects of Ferric Sulfate (Fe_2_(SO_4_)_3_) on the Removal of Cyanobacteria and Cyanotoxins: A Mesocosm Experiment

**DOI:** 10.3390/toxins13110753

**Published:** 2021-10-23

**Authors:** Kim Thien Nguyen Le, Eyerusalem Goitom, Hana Trigui, Sébastien Sauvé, Michèle Prévost, Sarah Dorner

**Affiliations:** 1Department of Civil, Geological and Mining Engineering, Polytechnique de Montréal, Montréal, QC H3C 3A7, Canada; eyerusalem.goitom@polymtl.ca (E.G.); hana.trigui@polymtl.ca (H.T.); michele.prevost@polymtl.ca (M.P.); sarah.dorner@polymtl.ca (S.D.); 2Department of Chemistry, University of Montréal, Montréal, QC H3C 3J7, Canada; sebastien.sauve@umontreal.ca; 3NSERC Industrial Chair on Drinking Water, Department of Civil, Geological and Mining Engineering, Polytechnique de Montréal, Montréal, QC H3C 3A7, Canada

**Keywords:** coagulation, cyanobacteria, cyanobacterial blooms, cyanotoxins, ferric sulfate, mesocosms, microcystins, water treatment

## Abstract

Cyanobacterial blooms are a global concern. Chemical coagulants are used in water treatment to remove contaminants from the water column and could potentially be used in lakes and reservoirs. The aims of this study was to: 1) assess the efficiency of ferric sulfate (Fe_2_(SO_4_)_3_) coagulant in removing harmful cyanobacterial cells from lake water with cyanobacterial blooms on a short time scale, 2) determine whether some species of cyanobacteria can be selectively removed, and 3) determine the differential impact of coagulants on intra- and extra-cellular toxins. Our main results are: *(i)* more than 96% and 51% of total cyanobacterial cells were removed in mesocosms with applied doses of 35 mgFe/L and 20 mgFe/L, respectively. Significant differences in removing total cyanobacterial cells and several dominant cyanobacteria species were observed between the two applied doses; *(ii)* twelve microcystins, anatotoxin-a (ANA-a), cylindrospermopsin (CYN), anabaenopeptin A (APA) and anabaenopeptin B (APB) were identified. Ferric sulfate effectively removed the total intracellular microcystins (greater than 97% for both applied doses). Significant removal of extracellular toxins was not observed after coagulation with both doses. Indeed, the occasional increase in extracellular toxin concentration may be related to cells lysis during the coagulation process. No significant differential impact of dosages on intra- and extra-cellular toxin removal was observed which could be relevant to source water applications where optimal dosing is difficult to achieve.

## 1. Introduction

Cyanobacteria are well adapted to survive and proliferate in water bodies worldwide. Eutrophication and climate change are the main drivers of cyanobacterial blooms [1]. Many cyanobacterial genera are commonly associated with toxicity, including *Microcystis, Planktothrix, Dolichospermum, Cylindrospermopsis, Nodularia, and Schizotrix* [2]. Cyanotoxins are the byproducts of metabolite formation processes. They may appear within the cell (intracellular) or released into in the water column (extracellular). Cyanotoxins can be classified into several main groups: hepatotoxins (microcystins and nodularins), cytotoxins (cylindrospermopsins), neurotoxins (e.g., anatoxin-a), dermatotoxins (e.g., lyngbyatoxin), endotoxins (lipopolysaccharides) and other toxins [3,4]. Microcystins are the most commonly detected hepatotoxins associated with cyanobacterial blooms worldwide [5,6]. Several health impacts, such as gastroenteritis and cutaneous reactions, have been identified [7]. Moreover, they are one of the many potential causes of water quality degradation in aquatic systems, having potential consequences for biological communities [8]. 

Given the challenges that cyanobacteria pose to drinking water production, effective remediation practices and safety measures are needed to protect aquatic communities and human health. Coagulation is a widely used water treatment technology for effectively removing cyanobacteria and undissolved toxin compounds in drinking water treatment plants [9] and some of these compounds have been used to treat lakes affected by blooms. For instance, aluminium sulfate has been applied in the U.S.A. to reduce cyanobacterial biomass from water bodies and remove internal phosphate [10]. In 2008, Lürling and van Oosterhout [11] also used polyaluminium chloride (PAC) to remove cyanobacteria blooms of *Aphanizomenon flos-aquae* in Lake Rauwbraken (Berkel-Enschot, The Netherlands). In a eutrophic fishpond in the Czech Republic, a polyaluminium hydrochloride coagulant (PAX-18) was used to remove *Planktothrix agardhii* blooms [12]. However, at laboratory scales, several studies have shown that some conventional types of coagulants (for example polyaluminium chloride, aluminium sulfate, and ferric chloride) can cause cell lysis, resulting cell bound toxin release into the water during the coagulation process [13,14,15]. 

Among the conventional chemical coagulants, ferric based-coagulants (e.g., ferric chloride (FeCl_3_), ferric sulfate (Fe_2_(SO_4_)_3_)) have been used in natural water sources to effectively remove cyanobacterial cells [10,16]. Previous studies have demonstrated that ferric coagulants can also reduce phosphorus concentrations in the water column and in the sediment, reducing the internal phosphorus loading from sediment and subsequently inhibiting cyanobacteria growth [17]. Iron supplementation, on the other hand, was thought to increase the abundance of cyanobacteria in eutrophic environments [18] and it was also considered as one of the inducing factors of the microcystin synthesis [19]. 

To the best of our knowledge, no short term outdoor mesocosm experiments with fresh blooms have been conducted to investigate the removal cyanobacterial cells and their toxins by ferric sulfate coagulants. Differences in the removal of cyanobacterial species has been observed in drinking water treatment, and it is unclear whether coagulants will selectively remove particular species [20]. There is a need to investigate the differential removal of cyanobacterial species in the water column. Although long term remediation efforts are critical for mitigating harmful cyanobacterial blooms, specific uses such as drinking or recreational waters require immediate attention. Therefore, our research aims to investigate the feasibility of the coagulation process in an in-situ scale, specifically, in cyanobacteria dominated lakes. The main objectives of this study are to: 1) assess the efficiency of ferric sulfate coagulant in removing harmful cyanobacterial cells, and their related toxins from lake water with cyanobacterial blooms on a short time scale, 2) determine whether some species of cyanobacteria are selectively removed, and 3) determine the differential impact of coagulants on intra- and extra-cellular toxins.

## 2. Results and Discussions

### 2.1. Impact of Ferric Sulfate on Cyanobacterial Cell Counts

#### 2.1.1. Taxonomic Cyanobacterial Cell Counts and Composition

In the beginning of the experiment (T0), several genera such as *Dolichospermum* spp., *Microcystis* spp., *Aphanothece* spp., *Aphanocapsa* spp., *Aphanizomenon* spp., *Chroococcus* spp., *Coelosphaerium* spp., *Merismopedia* spp. and *Pseudanabaena* spp. were identified in Missisquoi Bay (MB) and Petit Lac Saint-François (PLSF) (Appendix A). The genera *Dolichospermum*, *Microcystis*, *Aphanizomenon* were found in both control and coagulated mesocosms at both studied sites and they accounted for more than 70% of total taxonomic cell counts (Appendix A). The initial average total taxonomic cell counts varied across the study sites and the experimental events, as shown in Figure 1. Most mesocosms at both MB and PLSF sites had average total taxonomic cell counts greater than 10^5^ cells/mL. On 13 August 2019, the total taxonomic cell counts were recorded with more than 10^7^ cells/mL in MB. These above-mentioned cyanobacteria genera frequently occured at both study sites and varied in time with wide ranges of taxonomic cell counts, with *Microcystis*, *Dolichospermum*, and *Aphanizomenon* being the most common genera [9,21,22,23]. 

Except for an incident in MB on 13 August 2019, *Microcystis* was a common genus in all of the mesocosms, with dominance or representative percentages as shown in Figure 2 and Appendix A. The relative cell abundance of *Microcystis* (as measured by total taxonomic cell counts) ranged from more than 16.3% in MB (10 September 2018) up to 61.5% in PLSF (24 July 2019). *Microcystis aeruginosa* (*M. aeruginosa*) was a dominant species in MB while *Microcystis flos-aquae* (*M. flos-aquae*) was a dominant species in PLSF (Appendix A). Similarly, *Dolichospermum* spp. was present in high relative abundance in almost all of our experiments with *Dolichospermum spiroides* (*D. spiroides*) most frequently observed in samples. Their relative abundance ranged from 3.16% in PLSF (24 July 2019) to more than 73% in PLSF (26 June 2019) (Appendix A). In terms of biovolume, *Dolichospermum* was the dominant genus in all events at both sites with the exception of the event on 13 August 2019 where the biovolume of *Aphanizomenon* was more important (Appendix A).

Interestingly, within MB (on 13 August 2019), *Aphanizomenon flos-aquae* was the dominant species (more than 86% of total cell counts and 87% of total cyanobacterial biovolume) in all mesocosms (Appendix A). Several studies have also indicated that *Aphanizomenon flos-aquae* was frequently identified as the dominant species in MB [20,24]. That *Aphanizomenon* genus became dominant during this sampling event is likely due to a large variability of dissolved phosphorus (DP). The DP concentration increased approximately 18 times in the water as compared to previous events (10 September 2018 and 24 September 2018) (Table 1). The historical total phosphorus (TP) monitoring data in Missisquoi Bay has also shown an increasing trend in TP concentrations in August and that additional TP could be from the sediment TP concentration at that time [25]. Increases in P-loading in water bodies can indeed cause the dominance of cyanobacteria in the phytoplankton community, especially for diazotrophic cyanobacteria such as *Aphanizomenon* spp. [26,27,28]. The *Aphanizomenon* genus might be adapted to environments with higher P concentrations having been sensitive to changes in ambient DP [29,30].

*Aphanothece clathrata* was also observed in some events at both sites (MB, 10 and 24 September 2018; PLSF 24 July and 5 August 2019) (Appendix A). Small numbers of the *Pseudanabaena* genus was found in MB in 2018 and it was not present at either MB or PLSF in 2019. The *Aphanocapsa* genus was occasionally observed in MB in 2018, while it appeared only once in a control mesocosm in PLSF on 24 July 2019 (Appendix A). Taxonomic cell counts of “Other” include the genera *Chroococcus*, *Merismopedia*, and *Coelosphaerium* were extremely small and were negligible compared to the dominant genera (Figure 2). “Other” genera accounted for less than 1% of total taxonomic cell counts as well as total cyanobacterial biovolume (Appendix A).

#### 2.1.2. Removal of Cyanobacterial Cells

In the control mesocosms, after two days of exposure (T48), the average total taxonomic cell counts did not decrease in all of the control mesocosms in MB in 2018 and PLSF in 2019, except for MB on 15 August 2019 (Figure 1). The cyanobacterial genus composition was similar to that observed at T0, as demonstrated in Appendix A. In the natural environment, cyanobacteria in high numbers may be maintained for up to a week or two [31,32]. For the event on 15 August 2019, total taxonomic cell counts suddenly decreased by more than 90% in the control mesocosm in MB (Figure 1), and *D. spiroides* became a dominant species in the control mesocosm accounting for 43.1% of total taxonomic cell counts, followed by *Aphanizomenon flos-aquae* (36.2%) (Appendix A). Since the total cell abundance in MB (13 August 2019) exceeded 10^7^ cells/mL at T0, the decomposition process would release CO_2_ and organic acids into the water body leading to a decrease in pH [30,33]. A decrease in pH (<7.1) (Appendix A) may cause growth suppression of *Aphanizomenon flos-aquae* in the control mesocosms [34]. Moreover, a decrease in genus *Aphanizomenon* may be caused by the appearence of *Synechococcus* in the blooms [35]. 

In contrast, the overall cell concentration at T48 in the ferric-sulfate-coagulated mesocosms decreased considerably in all coagulant treated mesocosms, dropping down from around 10^6^ or 10^5^ to 10^5^ or 10^3^ cells/mL (MB September 2018 and PLSF July and August 2019) or from 10^7^ or 10^6^ to 10^5^ or 10^4^ cells/mL (MB August 2019 and PLSF June 2019) (Figure 1). Mesocosms exposed to 35 mgFe/L coagulant had a total cell count reduction of more than 98% when compared to T0 (Appendix A). Mesocosms exposed to 20 mgFe/L showed a decrease in cell count ranging from 71% in MB on 26 September 2018 to more than 85% in other treated mesocosms, except for the PLSF (26 July 2019) mesocosm with the lowest decrease (approximately 51% reduction) (Appendix A).

During two study periods (2018 and 2019), taxonomic cell counts of *Dolichospermum* spp., *Microcystis* spp., *Aphanizomenon* spp., and *Pseudanabaena* spp. decreased in the mesocosms containing 20 mgFe/L and 35 mgFe/L at both sites (Figure 2). Other species (*Chroococcus*, *Merismopedia*, and *Coelosphaerium*) were removed to below detection limits for both doses in mesocosms in MB in 2018 and 2019; however, they were stable in the lower dose mesocosm and removed to below detection limits in the higher dose mesocosm in PLSF in 2019 (Figure 2). The dose of 35 mgFe/L was more effective than the dose of 20 mgFe/L for removing total taxonomic cell counts and individual dominant cyanobacteria at both study sites (Figure 1, Appendix A). In June and July 2019, in PLSF, *Aphanothece* spp. was increased and *Coelosphaerium* spp. were less removed after coagulation in mesocosms with the dose of 20 mgFe/L (Appendix A). 

Coagulation of cyanobacteria, especially mixed-species bloom samples, may be harder than other pollutants as they are living organisms and complex interactions exist among species. Most drinking water research has focused on coagulation process performance for total cyanobacterial cells rather than studying the efficiency of removing individual cyanobacterial species. However, in some cases, the less effectively removed cyanobacteria can impact drinking water quality after treatment. A previous study reported that *Aphanizomenon* cells were poorly coagulated in a drinking treatment plant and passed through water treatment [20]. The impact of cyanobacterial physical and chemical characteristics on coagulation processes has been studied, and it was found that variable surface charges or algogenic organic matter (AOM) may interfere and prevent agglomeration [36,37,38,39,40]. Moreover, some types of cell morphologies can prevent close contact of cells or cell liberation from flocs by cell motility [37,41,42]. Types and doses of coagulants also impact coagulation processes. Higher cell removal efficiency by using ferric coagulant in comparison to alum for the removal of *M. aeruginosa* has also been observed [43]. Flocs of ferric coagulant with a large size and high density could increase separation performance by sedimentation [44]. The pH values at studied sites ranged between 6 and 8 (Table 1, Appendix A) wherein cyanobacterial cells easily combine with ferric-based coagulants [40,45]. 

A considerable change in cyanobacterial species composition was detected in several events in the coagulated mesocosms after 48 h (Appendix A). *Aphanothece clathrata* became a dominant species in both coagulated mesocosms on 12 September 2018 in MB and in the higher dose mesocosm in July and August 2019 in PLSF. In contrast, there was negligible change in cyanobacterial relative abundance on 15 August 2019 in MB and on 28 June 2019 in PLSF. 

Application of coagulants for intense blooms (>10^5^ cells/mL) in lake or reservoir drinking water sources may help to prevent the accumulation and breakthrough of cyanobacteria cells and their related toxins [9]. However, coagulation in the water source has the potential to lead to lower cell counts at the water intake and reduce cyanobacterial cell buildup in the clarifier sludge bed [46]. Therefore, treatment plant processes need to be carefully adjusted during the algal bloom season. For example, preozonation of raw water before coagulation may also help cell removal during clarification [46]. 

Principal component analysis (PCA) was performed to evaluate the relationship between environmental conditions and cyanobacterial cell counts in MB and PLSF (Appendix A). No relationship was found between dominant genus and the environmental conditions in the beginning of study period at both sites. Overall, the PCA plot for PLSF showed that the genus *Aphanothece* was associated with total dissolved solid (TDS), pH, temperature, total nitrogen (TN), total phosphorus (TP) and total organic carbon (TOC) in mesocosm with dose of 35 mgFe/L but not 20 mgFe/L at T48. A relationship was observed between these above mentioned parameters with genera *Aphanothece* and *Dolichospermum* in mesocosms with doses of 20 and 35 mgFe/L at T48 in MB. 

### 2.2. Impact of Ferric Sulfate on Intracellular Cyanotoxins

#### 2.2.1. Intracellular Cyanotoxin Concentration and Composition

At the time of experiment (T0), twelve intracellular microcystins (MCs) including MC-RR, MC-YR, MC-HtyR, MC-LR, MC-HilR, MC-WR, MC-LA, MC-LY, MC-LW, MC-LF, [Asp^3^]MC-RR and [Asp^3^]MC-LR, anatoxin-a (ANA-a), cylindrospermopsin (CYN), and some rarely monitored cyanopeptides (anabaenopeptin A (AP-A) and anabaenopeptin B (AP-B)) were detected (Figure 3 and Appendix A). The concentration of total intracellular microcystins (intracellular ΣMCs) at T0 varied across the sites and study periods but was greater than 10^3^ ng/L with the highest quantity of more than 10^5^ ng/L documented in PLSF on 26 June 2019 (Figure 3). The initial concentration of intracellular AP-A was higher than 10^3^ ng/L in MB on 13 August 2019 (Figure 4). Several studies found intracellular toxins including MC-LR, MC-YR, MC-LY, MC-RR, MC-LW, MC-LA, CYN in MB, and MC-LR, MC-LA, [Asp^3^]MC-LR in PLSF [9,23,47]. 

Except for one sampling event in MB on 13 August 2019, MC-LR was frequently detected, accounting for at least 40% and up to 75% (count on total intracellular toxin concentration) in all mesocosms (Appendix A). While the percentage of MC-RR was between 17%–23% in all sampling events in MB, MC-LY accounted for around 20%–27% of total intracellular toxin concentration in most PLSF sampling events. A small amount (less than 3%) of MC-LA was observed on 10 September 2018 and 13 August 2019, and up to 14% of total intracellular toxin concentration during 24 September 2018 sampling event in MB. AP-B was also detected in MB, ranging from 5% (10 September 2018) to 9% (13 August 2019) of total intracellular toxin concentration. In MB on 13 August 2019, the individual toxin with highest concentration was AP-A representing nearly 50% of total intracellular toxin concentration, followed by MC-LR and MC-RR (around 20%). Other MC’s relative abundance were negligible (Appendix A). 

Those detected were toxins associated with the cyanobacteria taxa at T0 in both MB and PLSF [2]. A significant correlation between intracellular microcystin and toxigenic genera such as *Dolichospermum* and *Microcystis* was found after sampling a series of 22 lakes in southern Québec [48]. Previous studies indicated that the maximum intracellular cyanotoxins concentration was related to amounts of potentially toxic cyanobacterial increase sing in the water body, which is consistent with our findings [21]. For instance, the highest intracellular toxin ΣMCs, which occurred during the event on 26 June 2019, coincided with the predominance of *Microcystis* and *Dolichospermum* and the highest total taxonomic cell counts. The opposite trend was observed on 05 August 2019, where the total taxonomic counts were the lowest. 

#### 2.2.2. Intracellular Cyanotoxin Removal

In the control mesocosms, the concentration of intracellular ΣMCs did not change considerably after 2 days except for MB on 13–15 August 2019, where its concentration was reduced up to 90% (Figure 3). Furthermore, the composition of cyanobacterial toxins including MC-LR, MC-RR, MC-LY and AP-A remained stable after two days (Appendix A). In MB in August 2019, AP-A concentrations also decreased by 90%, although it remained the toxin with the highest concentration, followed by MC-LR (14% of total intracellular toxin concentration) and MC-RR (12%). A reduction in the concentration of intracellular ΣMCs and AP-A in MB was probably due to a decrease in total taxonomic cell counts.

MCs retained in intact cells may persist for several months and may increase under conditions which are most favorable for their growth [49]. Several studies have indicated that the dominance of toxigenic strains and the subsequent increase in microcystins production were affected by environmental factors, such as temperature, light, intensity, pH, nutrient concentration (NO^−^, NH_4_^+^, PO_4_^3−^, Fe^2+^), salinity, CO_2_ concentration, and turbidity [49,50,51].

On the contrary, in all mesocosms with ferric sulfate addition, the concentration of intracellular ΣMCs remarkedly decreased by more than 97% for the two applied doses (Figure 3 and Appendix A). Ferric sulfate is also efficient in removing individual microcystins such as MC-LR, MC-LY, MC-RR and MC-LA (Figure 4 and Appendix A). In that table, we also observed a remarkable reduction in the concentration of intracellular AP-A (80%–99% of reduction) and AP-B (90%–99% of reduction) in MB. From a statistical perspective, there is no significant difference in removing intracellular ΣMCs and individual intracellular cyanotoxins between two applied dosages (Appendix A).

A coagulation process is one of the most important steps for the removal of cyanotoxins because the majority of cyanobacterial toxins naturally exist intracellularly and are retained within the cell [49]. However, several toxins are frequently found extracellularly (i.e., cylindropermopsin) in lake water or culture medium [52,53]. Our results for intracellular toxins may also include toxins adsorbed into particulate matter and the cells themselves in the water column. Microcystins toxin can be retained on natural suspended solids in rivers and reservoirs [49]. Several studies have indicated conventional coagulation is effective (up to 98%) for removing intracellular microcystin at the lab-scale [54,55]. 

### 2.3. Effect of Ferric Sulfate on Extracellular Cyanotoxins

#### 2.3.1. Extracellular Cyanotoxin Concentration and Composition

The concentration of extracellular ΣMCs at the time of experiment may vary within the same location. For instance, at the MB experiment site, an extremely high concentration of extracellular ΣMCs, around 115 μg/L, was observed in July 2010 [9] while lower concentrations, ranging from 50 to 191 ng/L, were recorded from June to September 2017 [22]. In PLSF, the concentration of ΣMCs was high, approximately 1.7 μg/L, in June 2010 and lower, around 35 ng/L, in the summer 2017 [23]. In our research, the initial concentration of extracellular ΣMCs was higher than 10^2^ ng/L with highest value (more than 10^3^ ng/L) documented in PLSF on 26 June 2019 (Figure 5). We also observed an initial extracellular concentration of more than 10^2^ ng/L of AP-A in MB on 13 August 2019 (Figure 6).

In Appendix A we describe the percentages of all extracellular cyanotoxins observed in our experiments. In control and coagulated mesocosms, while MC-LR was dominant at the PLSF site, representing 75%–90% of total extracellular toxin concentration in almost all events in PLSF, it was also dominant or significant at the MB site, accounting for at least 11% (10 September 2018) and up to 37% (24 September 2018) of the total extracellular toxin concentration. CYN toxin accounted for 40% to 67% (10 September 2018) and around 12%–55% (13 August 2019) of the total toxin concentration. In addition, relative abundance of AP-A toxin accounted for around 13% in MB on 24 September 2018 while it occasionally appeared in the control mesocosms on 12 September 2018 and 13 August 2019. 

#### 2.3.2. Extracellular Cyanotoxins Removal

After 48 h, in the control mesocosms, we observed a trend of either a slight increase in the concentration of extracellular ΣMCs in MB (2018 and 2019), or a negligible change in PLSF (2019) (Figure 5). Concentrations of individual cyanotoxins after 48 h are presented in the Figure 6. The change in the relative abundance of individual cyanotoxins is documented in Appendix A. In MB, the relative abundance of AP-A remained unchanged on 12 September 2018, but markedly decreased in two other experiments (26 September 2018, and 15 August 2019). In term of relative abundance, extracellular CYN decreased from 67% to 35.5% on 12 September 2018. An increase in the relative abundance of MC-LR and MC-RR was observed in MB in 2018 and 2019. In PLSF, the relative abundance of MC-LR decreased remarkably while MC-LY increased highly on 26 July 2019 (Appendix A). Overall, MC-LR remained as a representative toxin in MB and PLSF.

The stability of extracellular toxin concentration over a wide range of temperatures and pH in water has been observed [56,57,58,59]. For instance, a previous study examined the stability of variant MCs, ANA-a, CYN, AP-A and AP-B during short-term storage at the lab-scale by inoculating a mixture of these toxins into surface water with no detectable cyanotoxin levels collected in PLSF and samples were stored at ambient temperate (20 °C). Results of that paper showed that extracellular toxins MC-LR, CYN and ANA-a changed negligibly while MC-LA, MC-LY, MC-RR, AP-A and AP-B slightly decreased in surface water after two days [60]. Moreover, it also indicated that microcystins degraded more slowly in mixtures compared to individual toxins [61]. 

In the coagulated mesocosms, an increase in the concentration of extracellular ΣMCs was observed at both MB (26 September 2018, and 15 August 2019) and PLSF (28 June 2019) in coagulated mesocosms (Figure 5). Besides, AP-A, AP-B and CYN toxins also increased after 48 h (Figure 6). The increase in the concentration of the total extracellular toxins in coagulated mesocosms after treatment was likely due to cell lysis and desorption of cyanotoxins from sediments or particulate matter. Several laboratory experiments have shown that following coagulation, cyanobacterial cells are lysed, and their toxins are released into the water [62,63,64,65]. Moreover, extracellular cyanotoxins were found not just in the water column but also in sediments and other organic debris [61,66]. As a result, cyanotoxins may desorb and be released into aqueous water in these locations where toxins accumulate, causing their concentration in water to rise [66,67]. 

In contrast, a decreasing trend of concentration of extracellular ΣMCs was observed in MB (12 September 2018) and PLSF (26 July and 7 August 2019) (Figure 5) for coagulated mesocosms which could be related to biodegradation and adsorption [68]. However, in this study, biodegradation was not observed in the control mesocosms. In this case, the reduction in extracellular ΣMCs is probably more associated to adsorption on to the sediment at the bottom of the coagulated mesocosms. It was discovered that cyanotoxins including CYN, MC-RR, MC-LF and MC-LW showed high adsorption (36.4% and 72.6%) in sandy sediments [66]. There is no significant difference between control and coagulated mesocosms as well as two applied doses (20 and 35 mgFe/L) on removing extracellular toxins (Appendix A). 

## 3. Conclusions


*Dolichospermum*, *Microcystis*, and *Aphanizomenon* were the dominant genera throughout the experimental period. *Aphanothece* spp. and *Coelosphaerium* spp. were also documented at some events;Total cyanobacterial cells were efficiently removed, with more than 96% and 51% removal in mesocosms with applied dose of 35 mgFe/L and 20 mgFe/L, respectively. Significant differences in removing total cyanobacterial cells and several dominant cyanobacterial genera were observed between applied doses;Both applied doses of ferric sulfate help to efficiently remove *Dolichospermum*, *Microcystis*, and *Aphanizomenon. Aphanothece* and *Coelosphaerium* had a lower removal rate in mesocosms with a dose of 20 mgFe/L in PLSF;Intracellular microcystins (MC-RR, MC-YR, MC-HtyR, MC-LR, MC-HilR, MC-WR, MC-LA, MC-LY, MC-LW, MC-LF, [Asp^3^]MC-RR and [Asp^3^]MC-LR), anatoxin-a (ANA-a), cylindrospermopsin (CYN), anabaenopeptin A (AP-A) and anabaenopeptin B (AP-B) were detected throughout the experiment;Ferric sulfate effectively removes detected intracellular cyanotoxins but not extracellular cyanotoxins. More than 97% removal of total intracellular microcystins were achieved for both applied doses;Different dosages of ferric sulfate have almost the same effectiveness in removing intra- and extra-cellular cyanotoxins meaning that source water treatment will not be highly sensitive to suboptimal dosing.


## 4. Material and Methods

### 4.1. Study Site Description 

Petit Lac St. François (PLSF) and Missisquoi Bay (MB) were selected as study locations because they are both eutrophic lakes that are prone to cyanobacterial blooms during growing season. PLSF is located in south-western Quebec (Canada; 45°32′16′′ N; 72°02′11′′ W; max depth (Z) = 1.8 m). PLSF is relatively small, shallow lake with approximately 50% agricultural land use in its catchment. Phosphorus and dissolved organic carbon concentrations were found to be high [23]. MB is a shallow embayment of Lake Champlain, situated in southern part of Quebec (Canada; 45°02′23′′ N; 73°04′41′′ W; depth (Z) = 4.75 m). This site was chosen because of frequent occurrence of intense cyanobacterial blooms [21,22]. The presence of cyanobacteria in this bay poses a threat to drinking water treatment and recreational activities [9,20]. 

### 4.2. Mesocosms Experiments Description 

The mesocosm experiments were conducted over 6 different cyanobacterial bloom events, September 2018 and from June to August 2019 (Table 1) and each experiment lasted for 48 h. The mesocosm design followed the study of Wood [69], including the series of treatments and controls as described below. Cylindrical low-density polyethylene bags of 21 L mesocosms were suspended by floats on the shore of Lake Champlain and PLSF, Québec, Canada. Surface lake water and cyanobacteria harvested from scums at the field site were pooled and mixed thoroughly in a large (200 L) container to achieve as homogeneous starting mixture as possible. To conduct the experiment, 6 mesocosms were sets of duplicates, then filled with that mixed water of 21L and were left unamended to serve as a control (control mesocosm) or amended with 2 doses of the ferric sulfate coagulant: 20 mgFe/L (mesocosm 20) and 35 mgFe/L (mesocosm 35). Samples were mixed immediately after addition of coagulant. The mesocosms floated at the surface and were not covered. All of the environmental conditions are presented in Table 1. 

### 4.3. Preparation and Application of Chemical Coagulant 

Standard liquid grade ferric sulfate Fe_2_(SO_4_)_3_ with Fe content 12.2% was purchased from Kemira Water Solution, Inc. (Canada). Two different doses of ferric sulfate coagulant (20 mgFe/L and 35 mgFe/L) were administered in a mesocosms. The dosages were chosen based on previous study [20]. 

### 4.4. Sampling and Filtration Procedure 

Sterilized high density 1-L polypropylene bottles were used to collect water samples from center of each mesocosm before adding the coagulant (T0) and after the addition of the coagulant at 48 h (T48). Water in mesocosm was mixed thoroughly before pouring into sterilized 1-L bottles at T0 while mixing was avoided to take the supernatant water at T48. For taxonomic samples, water samples were taken in 40 mL glass vials and preserved in Lugol’s iodine solution. Subsamples were taken for cyanotoxin, total organic carbon (TOC), dissolved organic carbon (DOC) and specific nutrients: total nitrogen (TN), total phosphorus (TP), dissolved nitrogen (DN), dissolved phosphorus (DP).

Subsamples of 120 mL were filtered on hydrophilic polypropylene GHP, 0.45 µm filters (PALL, Mississauga, ON, Canada). After filtration, the filter for intracellular toxin (sorbed to particulate matter or cell-bound) was placed in a petri dish (Millipore Sigma, Oakville, ON, Canada) while the filtrate for extracellular toxin (dissolved toxin) was transferred into 125 mL amber Nalgene™ polyethylene terephthalate glycol (PETG) bottles (Thermo Scientific, Mississauga, ON, Canada). Samples for TOC, TN, and TP were filled directly. The DOC samples were filtered by a pre-rinsed (1 L MilliQ water) 0.45 µm membrane filters (PALL, Mississauga, ON, Canada). A filter membrane 0.45 μm, 47 mm (Millipore Sigma, Oakville, ON, Canada) was used for subsample of DN/DP. Toxin and nutrient samples were taken in duplicate. 

Toxin followed by storage at −25 °C. TN/TP/TOC/DOC samples were stored at 4 °C until analysis. Taxonomic samples were stored in a dark place at room temperature. 

### 4.5. Analysis Methods

Taxonomic identification was achieved by using inverted microscopy in a Sedgwick-Rafter chamber with 40× magnification (Leica Microsystems GmbH, Wetzlar, Germany) [20,70,71], at the Université du Québec à Montréal’s (UQAM) Biological Sciences Department.

A total of 17 cyanobacterial toxins were tested with 12 microcystins (MC-RR, MC-YR, MC-HtyR, MC-LR, MC-HilR, MC-WR, MC-LA, MC-LY, MC-LW, MC-LF, [Asp^3^]MC-RR and [Asp^3^]MC-LR), anatoxin-a (ANA-a), cylindrospermopsin (CYN), anabaenopeptin A (AP-A) and anabaenopeptin B (AP-B). Total microcystins samples were detected via a Lemieux oxidation step and an online solid phase extraction (SPE) and salting coupled to ultrahigh performance liquid chromatography tandem mass spectrometry (UHPLC-MS/MS). The limit of detection was 0.5 ng/L and accuracy was in the range of 93%–110%, deemed satisfactory. An online solid phase extraction ultra-high performance liquid chromatography high-resolution mass spectrometry (SPE-UHPLC-HRMS) was developed for those multi-toxins with a limit of detection between 8 and 53 ng/L and recoveries ranging from 81% to 113%. More detail in [72,73]. 

The TOC and DOC were measured on a OI Instrument, Aurora 1030 according to method 415.1 [74]. TN and DN were analyzed on a Lachat, Quickchem 8500 based on the method 353.2 and the method 350.1 [75]. Standard method numbered 365.3 and 365.1 [76] were applied for phosphorus and phosphate with machine of Astoria-Pacific, Astoria 2. TOC/DOC and specific nutrient samples were analyzed at Groupe de Recherche Interuniversitaire en Limnologie, University of Montréal.

An online YSI EXO2 water-quality multi-probe (YSI, Yellow Springs, Greene County, OH, USA) was fitted with total algae sensor (chlorophyll-a/phycocyanin) was applied in this paper to measure phycocyanin, chlorophyll a, pH, total dissolved solid (TDS) and temperature [9,20].

### 4.6. Statistical Analysis

Statistical analysis was performed by R (R Core Team, 2020). All of the plots in this paper were generated by ggplot2 package in R (https://ggplot2.tidyverse.org, accessed on 12 March 2021). Principal component analysis (PCA) was performed to evaluate the impact of environmental conditions on total cell counts and cyanobacterial species in MB and PLSF, it is with the vegan (https://cran.r-project.org/package=vegan, accessed on 10 October 2021) and missMDA (https://cran.r-project.org/package=missMDA, accessed on 22 April 2021). Statistical tests were normalited by the Shapiro-Wilk test in mvnormtest package (https://cran.r-project.org/package=mvnormtest, accessed on 03 July 2021) and one-way ANOVA (Analysis of variance) test. Statistical significance was set at a 0.05 *p*-value cut-off (*p* < 0.05). Imputation of missing value is conducted by using the left-censored imputation method in MICE (https://cran.r-project.org/package=mice, accessed on 05 July 2021) and QGCOMP packages (https://cran.r-project.org/package=qgcomp, accessed on 20 June 2021).

## Figures and Tables

**Figure 1 toxins-13-00753-f001:**
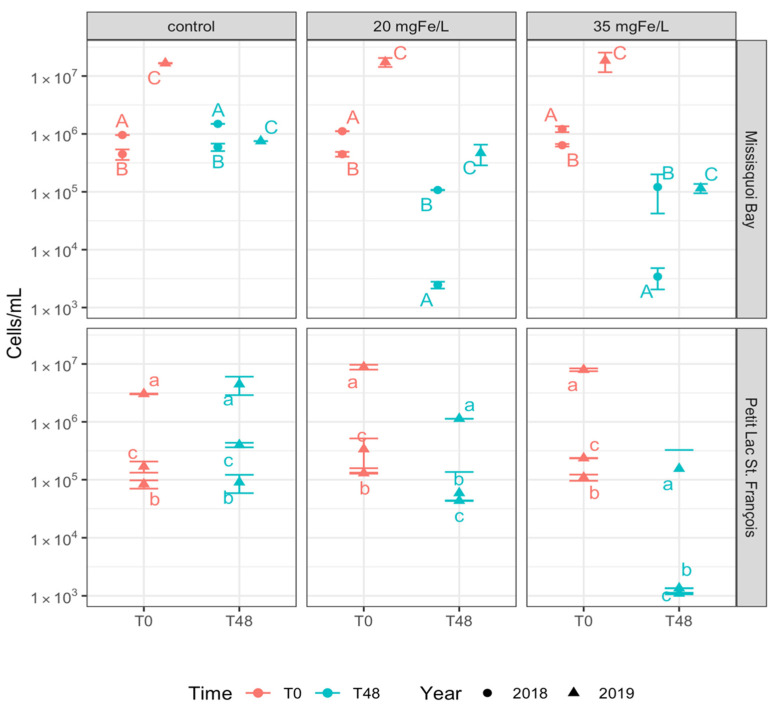
Total taxonomic cell counts in control, 20 mgFe/L, and 35 mgFe/L mesocosms (Mean ± standard deviation): A) 10–12 September 2018; B) 24–26 September 2018; C) 13–15 August 2019; a) 26–28 June 2019; b) 24–26 July 2019; c) 05–07 August 2019.

**Figure 2 toxins-13-00753-f002:**
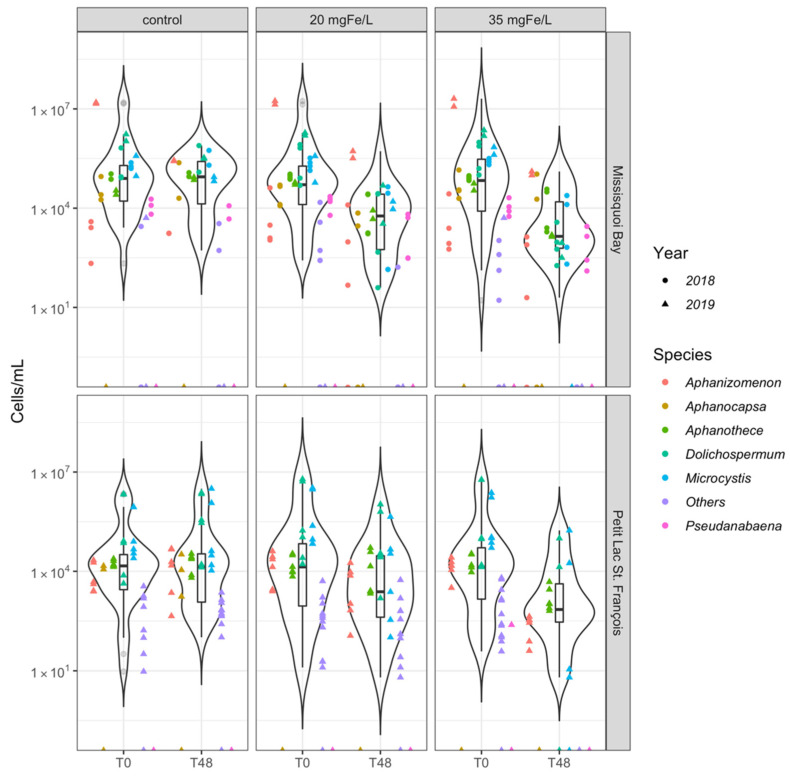
Violin plot shows the distribution of taxonomic cell counts at level genus in control, 20 mgFe/L, and 35 mgFe/L mesocosms. (Others include *Chroococcus*, *Coelosphaerium*, *Merismopedia*). The bottom and top of the box shows the lower and upper quartiles, the band in between them shows the median, whiskers show the minimum and maximum (excluding outliers) and circles show the outliers. Outliers are values more than 1.5 times the length of the interquartile range greater than the upper quartile or smaller than the lower quartile.

**Figure 3 toxins-13-00753-f003:**
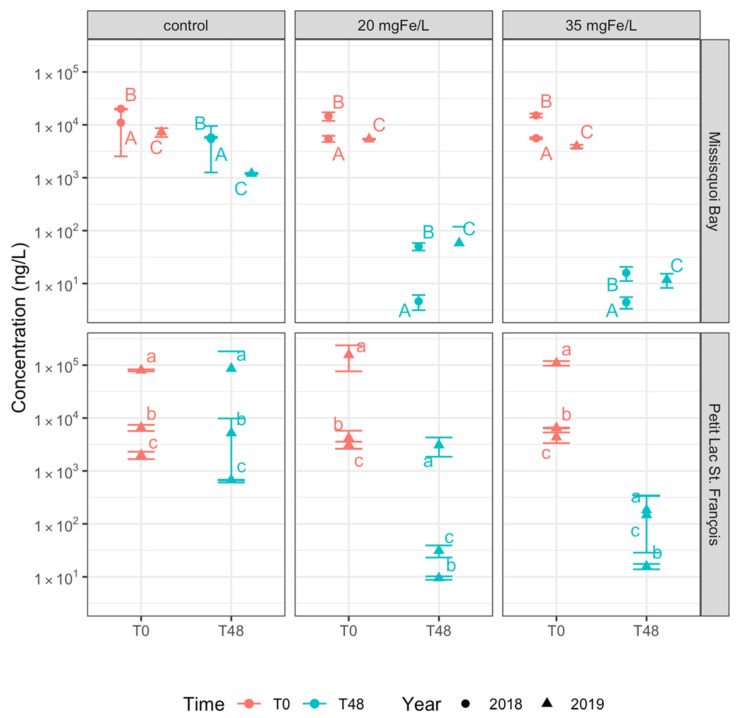
Concentration of total intracellular microcystins in control, 20 mgFe/L, and 35 mgFe/L mesocosms (Mean ± standard deviation): A) 10–12 September 2018; B) 24–26 September 2018; C) 13–15 August 2019; a) 26–28 June 2019; b) 24–26 July 2019; c) 05–07 August 2019.

**Figure 4 toxins-13-00753-f004:**
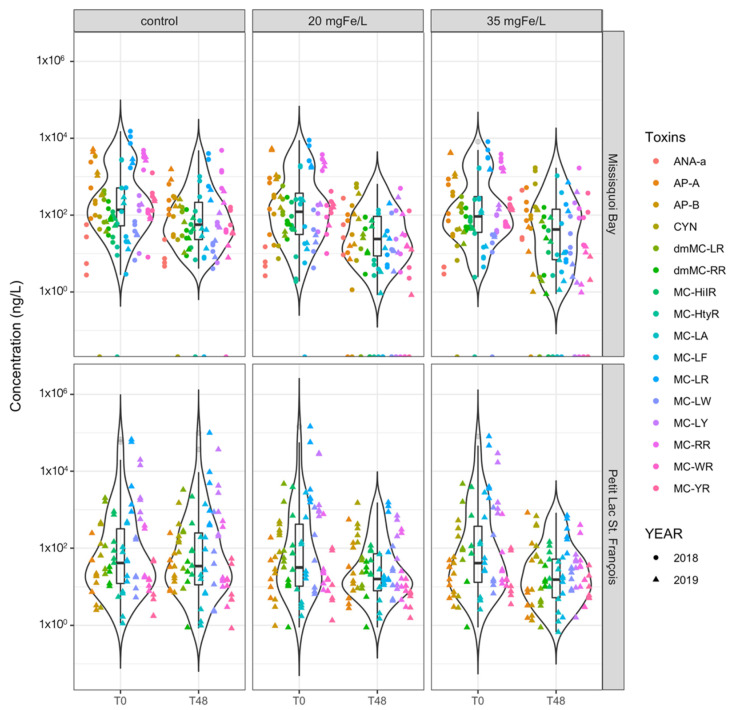
Violin lot shows the distribution of intracellular toxin concentration of individual cyanotoxins in control, 20 mgFe/L, and 35 mgFe/L mesocosms in Missisquoi Bay (MB) and Petit Lac St. Francois (PLSF). The bottom and top of the box show the lower and upper quartiles, the band in between them shows the median, whiskers show the minimum and maximum (excluding outliers) and circles show the outliers. Outliers are values more than 1.5 times the length of the interquartile range greater than the upper quartile or smaller than the lower quartile.

**Figure 5 toxins-13-00753-f005:**
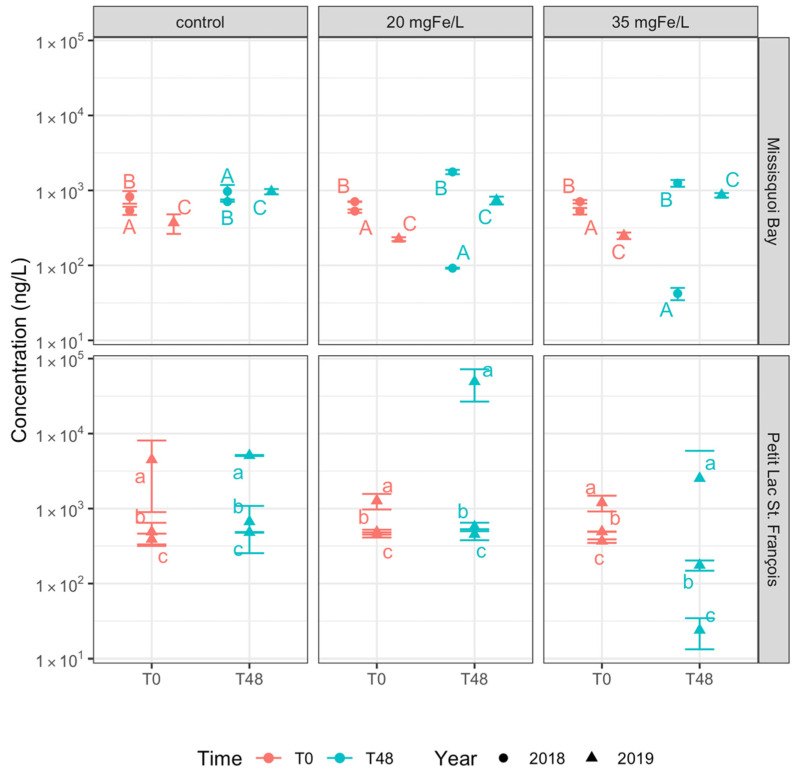
Concentration of total extracellular microcystins in control, 20 mgFe/L, and 35 mgFe/L mesocosms (Mean ± Standard deviation). A) 10–12 September 2018; B) 24–26 September 2018; C) 13–15 August 2019; a) 26–28 June 2019; b) 24–26 July 2019; c) 05–07 August 2019.

**Figure 6 toxins-13-00753-f006:**
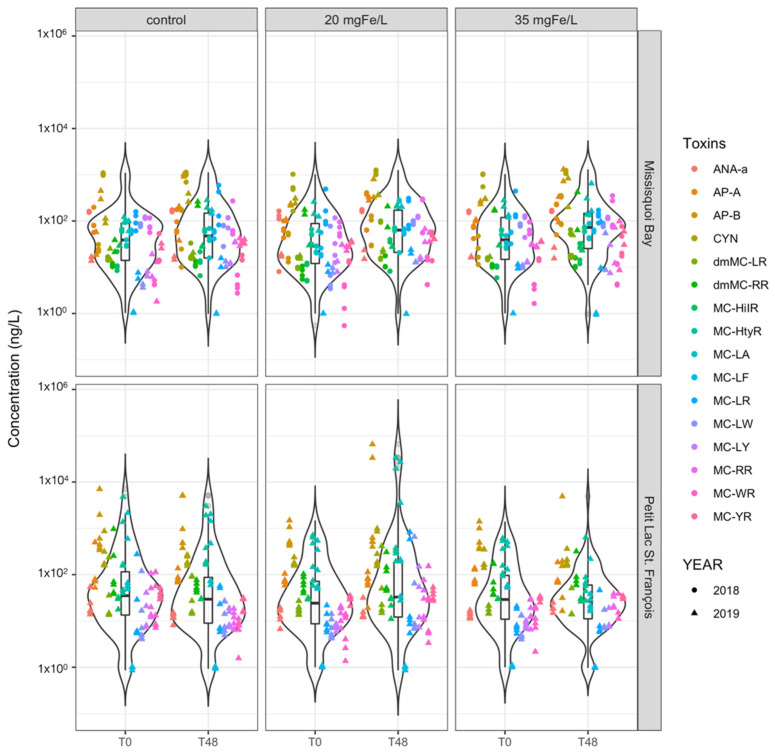
Violin plot shows the distribution of extracellular toxin concentration of individual cyanotoxins in control, 20 mgFe/L, and 35 mgFe/L in Missisquoi Bay (MB) and Petit Lac St. François (PLSF). The bottom and top of the box show the lower and upper quartiles, the band in between them shows the median, whiskers show the minimum and maximum (excluding outliers) and circles show the outliers. Outliers are values more than 1.5 times the length of the interquartile range greater than the upper quartile or smaller than the lower quartile.

**Table 1 toxins-13-00753-t001:** Environmental conditions of lake water samples in mesocosms (*n* = 6) at T0 (Mean ± standard deviation).

Parameters	Missisquoi Bay			Petit Lac St. François		
	(A) 10 September 2018	(B) 24 September 2018	(C) 13 August 2019	(a) 26 June 2019	(b) 24 July 2019	(c) 05 August 2019
Total cell counts (cells/mL)	998,183 ± 35,034	547,325 ± 16,578	17,408,158 ± 138,898	6,033,197 ± 316,425	109,193 ± 3578	235,723 ± 5986
Chlorophyll-*a* (RFU)	-	-	62.22 ± 1.03	3.97 ± 0.04	5.14 ± 0.06	6.27 ± 0.08
Phycocyanin (RFU)	-	-	93.43 ± 0.52	16.51 ± 1.54	1.77 ± 0.04	4.83 ± 0.13
pH	6.5 ± 0.08	6.4 ± 0.07	8.05 ± 0.24	8.08 ± 0.07	7.84 ± 0.03	8.01 ± 0.01
TDS (mg/L)	101 ± 0.00	100 ± 0.00	98.00 ± 0.00	122.50 ± 0.71	116.00 ± 0.00	115.00 ± 0.00
Temp (°C)	21.8 ± 0.01	18.7 ± 0.12	26.89 ± 0.21	25.96 ± 0.49	25.14 ± 0.12	24.43 ± 0.24
TOC (mg C/L)	15.22 ± 0.25	5.55 ± 0.07	885.00 ± 33.34	175.00 ± 9.50	11.10 ± 0.38	10.76 ± 0.13
DOC (mg C/L)	7.50 ± 0.05	5.00 ± 0.00	19.34 ± 1.67	9.80 ± 0.11	9.83 ± 0.13	11.34 ± 1.58
TN (mg N/L)	5.55 ± 0.65	2.75 ± 0.08	7.84 ± 2.95	12.85 ± 0.57	1.11 ± 0.05	1.29 ± 0.008
TP (μg P/L)	360.51 ± 0.01	292.11 ± 13.28	4336.92 ± 74.65	723.55 ± 24.97	110.38 ± 1.89	131.78 ± 7.71
DN (mg N/L)	0.45 ± 0.004	0.52 ± 0.009	2.66 ± 0.12	0.73 ± 0.08	0.60 ± 0.02	0.50 ± 0.02
DP (μg P/L)	17.05 ± 0.07	17.99 ± 0.45	302.17 ± 12.98	139.06 ± 3.35	23.92 ± 4.91	42.86 ± 1.13

## Data Availability

Data is contained within the article and Appendix A.

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
