# Peer review of "The Effects of Ferric Sulfate (Fe2(SO4)3) on the Removal of Cyanobacteria and Cyanotoxins: A Mesocosm Experiment"

_toxins, 2021, doi:10.3390/toxins13110753_

Round 1
Reviewer 1 Report
Huge data, including the taxonomy and several kinds of cyanotoxins, need to be discussed in this study. Thus, statistical analysis is a good idea to understand the effects of ferric sulfate. However, the authors just run the ANOVA in the removal of intracellular and extracellular cyanotoxins to test the differences in the concentration of ferric sulfate. The result shows no significant difference between Mesocosm 20 and 35 mgFe/L (p-value > 0.05, Tables S3 and S4), however, the authors didn’t conclude reliable information for the effect of the concentration of ferric sulfate either. Therefore, more statistical analysis was required in some issues for better understanding: e.g. (1) For the total taxonomic cell counts, figure out the difference in removal efficiency based on the concentration of ferric sulfate; (2) it is hard to understand the removal rate of cyanotoxin concentration in each sampling period based on Figures 2, 4, and 6. However, the relative abundance of cyanotoxin in Figures S1 to S4 can not figure out the removal rate either. it will be better if the authors consider the removal efficiency of each cyanotoxin based on ferric sulfate addition by running the statistical analysis. In another hand, several water quality parameters were collected in this study, it will be better if the authors figure out the relationship between water quality and taxonomic (or cyanotoxins) based on PCA results.
A few comments are listed below. Please confirm.
Keywords
Some keywords were not appropriate, please rephrase them, e.g. eutrophication, extracellular, intracellular.
Results and Discussions
Line 81-85: The authors use “sp.” to represent all species under a genus. However, the “sp.” is a species’ name that needs to be written in Italic. For authors' purpose, please use “spp.” (without Italic) instead of “sp.”. Please double-check the remaining text for the same issue.
Line 90: Please change “cell/mL” to “cells/mL”.
Figure 1: Please change “cell/mL” of the Y axis to “cells/mL”.
Line 211: Please use “MC-” for all analogs of microcystins which been detected in this manuscript and added the full name of dmMC-RR and dmMC-LR.
Line 214: Both ΣMC and ΣMCs appeared in this manuscript, please unify the written in the remaining text.
Figure 2, 4, and 6: Please add the meaning of the box plot and the curve in the caption. In addition, the name of genera in Figure 2 should be written in Italic.
Line 345-346: This sentence seems like come from literature instead of this study, please add the citation.
Line 349-351: The authors mention that the decrease of extracellular microcystins could be related to biodegradation. However, there no biodegradation was observed in the coutorl experiment. Thus, it is not a good idea to link the decrease to biodegradation.
Conclusions
Line 374-375: Please unison the abbreviation of cyanotoxin with previous text.
Material and Methods
Section 4.6: Please remove the sentence of PCA analysis because there was no PCA result shown in the manuscript.
Author Response
Dear Reviewer,
We would like to thank you for your extremely useful and constructive comments for a better version of our manuscript. We attach here the response to all your comments.
We would like to thank you again for your valuable feedback and your time.
With kind regards

Reviewer 2 Report
The study is well conducted and brings interesting data. Its novelty lies in using natural water samples with various cyanobacteria and toxin composition. I recommend the study for publishing after revisions. Specific comments are given below.
Abstract: Please state more clearly that the study deals with coagulation in water reservoirs and not in water treatment plant.
lines 12-14 „while using ferric sulfate during coagulation is efficient in removing cyanobacterial cell (more than 71%), higher dosages are proved to be more efficient in term of both total cells as well as individual cyanobacteria removal.“ I recommend being more specific, give numbers – state, what dosages were used with what efficiencies. It is not clear that high efficiencies were reached with higher dosage. You can also state the initial cell concentrations.
lines 17-18 Can you also be more specific (give numbers)?
lines 18-19 Please state why extracellular toxin concentrations increased.
line 19 „No significant differential impact of dosages is observed“ please be more clear. Do you mean impact on toxin removal or cyanobacteria removal or both?
line 148 “decrease” instead of “decreased”
lines 184-185 “Flocs of ferric coagulant with a large size and high density could increase the coagulation process performance.” I would rather say that these flocs increase separation performance by sedimentation.
lines 185-188 “The pH values at studied sites ranged between 6 and 8 (Table 1) wherein cyanobacterial cells easily combine with ferric-based coagulants [46,50,53,55]; those pH and selected doses would result in precipitation of ferric (III) hydroxyte in which cells become enmeshed and were thus removed.” The first part of the sentence is right, but this also depends on pH value after adding coagulant, which may be much lower than initial pH values as the dosages of Fe are relatively high. pH lowering after addition of coagulant depends on (and can be estimated from) water sample neutralizing capacity (alkalinity). The second part of the sentence is a speculation, it is hard to say. It is usually difficult to distinguish between enmeshment and other coagulation mechanisms without thorough examination of particular coagulation process. Enmeshment occur at higher pH values (which is probably not this case, because pH probably drops), where the formation of ferric hydroxyte is fast. Moreover, I cannot imagine how enmeshment occurs without mixing the sample. I assume that charge neutralization and adsorption onto ferric hydroxyte precipitates are more probable coagulation mechanisms in this case.
lines 189-200 This text should be rearranged. I do not see linkage between these two sentences: “There was negligible change in cyanobacterial relative abundance on August 15, 2019 in MB and on June 28, 2019 at PLSF. Application of coagulants for intense blooms (>105 cells/mL) in lake or reservoir drinking water sources may help to prevent the accumulation and breakthough of cyanobacteria cells and their related toxins.”
lines 251-252 “Furthermore, the composition of cyanobacterial toxins including MC-LR, MC-RR, MC-LY and AP-A remained stable after coagulation (Figure S3).” I would move this information to the text about mesocosms with coagulation. It does not fit in the first paragraph, which is about control mesocosms.
lines 273-274 “the majority of cyanobacterial toxins naturally exist intracellularly and are retained within the cell” Can you quantify in this study how the concentrations of intracellular and extracellular toxins differ? It would be interesting.
line 307 Information about the coagulated mesocosms does not fit into this paragraph.
Conclusions – I would appreciate giving numbers about efficiency and maybe also the initial cell concentrations. “Different dosages of ferric sulfate have almost the same effectiveness in removing intra- and extracellular cyanotoxins meaning that source water treatment will not be highly sensitive to suboptimal dosing.” It depends on what is the optimum dosage (or how wide is the interval of optimum dosages) and how far are the used dosages from optimum.
line 390 „intensecyanobacterial“ should be „intense cyanobacterial“
line 404 How these coagulant dosages were chosen?
line 419 Please explain abbreviation GHP.
In section „4.2. Mesocosms Experiments Description“ I miss some information about the experiment: Were the samples mixed after coagulant dosing? At what conditions were the mesocosms stored (temperature, light)?
Adding cell counts to Table 1 may be useful.
Author Response

(The authors gave the same response as above.)

Reviewer 3 Report
This paper investigated the removal of cyanobacterial cells and 65 their toxins by ferric sulfate coagulants based on a short term outdoor mesocosm experiments with fresh blooms. This work based on traditional ferric salt coagulation to remove algal pollutants has strong practical significance. The result has certain reference value for the optimization and improvement of the enhanced flocculation process of other eutrophic lake water. This study was well organized and could be considered after minor revision. Please consider the following comments:
- Line 167-168, it was mentioned that Aphanothece and Coelosphaerium sp. were repeatedly poorly removed or even increased in numbers after treatment in mesocosms dosed with 20 mgFe/L at both sites. According to Table S2, the removals of Aphanothece sp. and Coelosphaerium sp. were lower at sampling sites of PLSF June and July. At MB sites the removals were over 60% for both algae. The description of ‘poorly removed at both sites’ should be reconsidered. Meanwhile, due to the different algae removal efficiencies between June, July and other months, the water quality of PLSF in June and July should be discussed together.
- Line 220-222, except for one sampling event at MB on August 13, 2019, MC-LR was frequently detected, accounting for at least 40% and up to 75% (count on total intracellular toxin concentration) (Figure S3). For each sampling site, the concentration of MC-LR is dominant. The relationship between the type and quantity of algae at each site and the concentration of intracellular toxin needs to be further clarified. By observing the T0 group in Figure S1 and S2, the species of algae changed with the sampling time, and the correlation with the concentration distribution of algae toxins could be discussed.
- Some note numbers and data points stacking in figures 1, 3 and 5 need to be adjusted.
- Some typos were in the manuscript, such as at line 236, ‘the authors of [59] found’, at line 306, ‘AP-A was only high in in the control mesocosm…’, the manuscript should be checked carefully.
- The biovolume was described as (>103 mm3/L) at line 242, but it was described as (approximately 40 cells/mL) at line 244. The description of this indicator should better keep consistent.
- As a research article, there are too many references in this manuscript and some of the citations need to be updated.
- Is the data of lake water samples in mesocosm at T48 available in the experiments (especially for the final pH)?
Author Response

(The authors gave the same response as above.)

Round 2
Reviewer 1 Report
This manuscript can be accepted for publication